# OpenReview forum: "HaluEval: A Large-Scale Hallucination Evaluation Benchmark for Large Language Models"
_EMNLP/2023/Conference — EMNLP 2023 Main_

### Official Review · Reviewer_ANev · 2023-07-31

**Soundness:** 4

**Excitement:**

3: Ambivalent: It has merits (e.g., it reports state-of-the-art results, the idea is nice), but there are key weaknesses (e.g., it describes incremental work), and it can significantly benefit from another round of revision. However, I won't object to accepting it if my co-reviewers champion it.

**Paper Topic And Main Contributions:**

The paper proposed a Hallucination Evaluation for LLMs (HaluEval) benchmark, a large collection of generated and human-annotated hallucinated samples for evaluating the performance of LLMs in recognizing hallucination. Based on the HaluEval benchmark, the paper tested LLM-based Hallucination Recognition methods and tried three improvement strategies (Knowledge Retrieval, CoT Reasoning and Sample Contrast), results shows that Knowledge Retrieval and CoT Reasoning can improve the ability of LLMs to recognize hallucination.

Main Contribution:
1. The paper constructed a hallucination dataset with a two-step pipeline and analyze what types of content and to which extent LLMs tend to hallucinate.
2. The paper tested LLMs as hallucination detector on the constructed dataset and further investages three improvement strategies.

**Reasons To Accept:**

1. The paper is well written, the methods are easy to understand and the contribution is clear.
2. The constructed dataset is large scale with multiple tasks, which is usefull for future work.
3. The fragments of the hallucinated content in the LLM responses is helpful for future work.

**Reasons To Reject:**

1. The paper investage the LLM hallucination issue, as shown in Table 1, the LLM is asked to generate a response given a normal user query. However, in the benchmark construction, the instruction includes some frames that force the LLM to generate hallucinated content such as "You should try your best to make the answer become hallucinated." and given some constrains such as "#Hallucinated Answer# can only have about 5 more words than #Right Answer#." In this way, the distribution of types and extent of hallucination may not be exact same as the hallucination that the LLM will generate when given a normal query as shown in Table 1.

**Reproducibility:**

4: Could mostly reproduce the results, but there may be some variation because of sample variance or minor variations in their interpretation of the protocol or method.

**Reviewer Confidence:**

4: Quite sure. I tried to check the important points carefully. It's unlikely, though conceivable, that I missed something that should affect my ratings.

---

> ### Author Rebuttal · Authors · 2023-08-29
>
> **Question 1: The paper investigates the LLM hallucination issue, as shown in Table 1, the LLM is asked to generate a response given a normal user query. However, in the benchmark construction, the instruction includes some frames that force the LLM to generate hallucinated content such as "You should try your best to make the answer become hallucinated." and given some constrains such as "#Hallucinated Answer# can only have about 5 more words than #Right Answer#." In this way, the distribution of types and extent of hallucination may not be exact same as the hallucination that the LLM will generate when given a normal query as shown in Table 1.**
>
> **Response to Reject 1:** Thanks for your comments. Many previous studies have found that the hallucinations in LLM responses are often mixed with correct content, but exhibit minor difference from the correct content [1][2], which are also discovered in our human annotated ChatGPT responses. This finding inspires us to intentionally prompt LLMs to generate hallucinations by mimicking the given right answer.  In our instruction, the force frames "You should try your best to make the answer become hallucinated." and the length constraint "#Hallucinated Answer# can only have about 5 more words than #Right Answer#." are necessary to make our synthesized examples closer to the distribution of right answers, which is designed based on our preliminary experiments. Without the force frame, we discovered that ChatGPT may repeat the original answer and does not generate hallucination examples. And without the length constraint, the synthesized hallucinations from ChatGPT will be much longer than the original answer. These two frames are added in order to make our synthesized data close to the distribution of right answers and cannot be trivially distinguished from the right answers.
>
> More importantly, we carefully design the instruction details, i.e., hallucination patterns and demonstrations (described in Line 152 Instruction Design), for guiding LLMs how to revise the right answer to make it hallucinated and closer to the distribution of real hallucinations LLMs usually generate. First, the hallucination patterns are task-specific and reflect the commonly present way where LLMs produce hallucinations [2][3][4]. These patterns can guarantee that the types of synthesized hallucinations are the same as in real task LLMs will produce. Second, the in-context hallucination demonstrations can further help LLMs understand these hallucination patterns and synthesize more accurate hallucinated examples according to the provided demonstrations.
>
> **References:**
>
> [1] Dziri, Nouha, Sivan Milton, Mo Yu, Osmar Zaiane, and Siva Reddy. "On the origin of hallucinations in conversational models: Is it the datasets or the models?." arXiv preprint arXiv:2204.07931 (2022).
>
> [2] Das, Souvik, Sougata Saha, and Rohini K. Srihari. "Diving Deep into Modes of Fact Hallucinations in Dialogue Systems." arXiv preprint arXiv:2301.04449 (2023).
>
> [3] Zheng, Shen, Jie Huang, and Kevin Chen-Chuan Chang. "Why does chatgpt fall short in providing truthful answers." ArXiv preprint, abs/2304.10513 (2023).
>
> [4] Cao, Meng, Yue Dong, and Jackie Chi Kit Cheung. "Hallucinated but factual! inspecting the factuality of hallucinations in abstractive summarization." arXiv preprint arXiv:2109.09784 (2021).

---

### Official Review · Reviewer_XvWz · 2023-08-05

**Soundness:** 5

**Excitement:**

4: Strong: This paper deepens the understanding of some phenomenon or lowers the barriers to an existing research direction.

**Paper Topic And Main Contributions:**

It is smooth to read this paper, which is about building benchmark dataset (HaluEval) for hallucination detection.

The data is collected in sample-then-filtering strategy. The sample part is done with chatgpt (Azure ChatGPT API is used). The filtering is relied on manual labeling. This leads to a big benchmark, including 35000 samples (regular+hallucinated), 5000 general user queries with responses and 30000 task specific examples in 3 major tasks (QA, Dialogue and Summarization).

Based on the collected benchmark, major LLMs are examined on the benchmark including ChatGPT and GPT-3. Analysis is also provided to give more insights on the experiment and analysis results.

It might be better if other LLMs can be tested on the benchmark. Or maybe a leaderboard can be provided to collect state-of-the-art performance on the dataset.


**Reasons To Accept:**

(1) large scale hallucination detection dataset, which is expected by the community

(2) the dataset covers different tasks (QA, Dialogue and Summarization), with detailed info about the hallucination


**Reasons To Reject:**



**Reproducibility:**

4: Could mostly reproduce the results, but there may be some variation because of sample variance or minor variations in their interpretation of the protocol or method.

**Reviewer Confidence:**

4: Quite sure. I tried to check the important points carefully. It's unlikely, though conceivable, that I missed something that should affect my ratings.

---

> ### Author Rebuttal · Authors · 2023-08-29
>
> **Question 1: It might be better if other LLMs can be tested on the benchmark. Or maybe a leaderboard can be provided to collect state-of-the-art performance on the dataset.**
>
> **Response to contributions:** We appreciate your valuable suggestions. We tested the recently emerged models after submission, including a close-source LLM (Claude-2) and three open-source LLMs (LLaMA 2 Chat 7B, ChatGLM 7B, and Falcon 7B). Their results are shown as below. Building a leaderboard is a good way to compare existing powerful LLMs and collect state-of-the-art performance in our dataset. We will release this leaderboard in our github page after the review period. Thanks for your precious comments again.
>
> |      | QA | Dialogue | Summarization | General|
> | ----------- | ----------- |----------- |----------- |----------- |
> | **Claude-2**|    69.78%  |   64.73%    |57.75%|75.00%|
> | **LLaMA-2 Chat (7B)** | 49.60%   | 43.99%        |49.55% |20.46%|
> | **ChatGLM (7B)** | 47.93%   | 44.41%        |48.57%|30.92%|
> | **Falcon (7B)** | 39.66%   | 29.08%        |42.71%|18.98%|

---

### Official Review · Reviewer_WrMd · 2023-08-05

**Soundness:** 4

**Excitement:**

3: Ambivalent: It has merits (e.g., it reports state-of-the-art results, the idea is nice), but there are key weaknesses (e.g., it describes incremental work), and it can significantly benefit from another round of revision. However, I won't object to accepting it if my co-reviewers champion it.

**Paper Topic And Main Contributions:**

The authors propose HaluEval, a benchmark for evaluating hallucinations generated by large language models in response to open-world queries. The dataset consists of hallucinations in response to queries from various sources (HotpotQA, OpenDialKG, and CNN-DailyMail), and recent instruction-tuned models are tested for their ability to detect generated hallucinations. The findings suggest that current models have weak detection capabilities, but knowledge retrieval augmentation can improve performance.

**Questions For The Authors:**

- Could the authors clarify why the benchmark is reasonable for hallucination detection, given that models were explicitly asked to generate hallucinated answers?
- As mentioned in the weakness, could the authors clarify the chain-of-thought setting?

**Reasons To Accept:**

- The benchmark addresses a very important issue of hallucination in large language models, and having benchmarks to enable new detection methods is also important
- The evaluation also indicates that current models are extremely weak at detecting hallucinations and calls for future work to improve hallucination detection.

**Reasons To Reject:**

- The datasets do not appear to fully align with the task at hand, specifically regarding the detection of hallucinations. Models have been intentionally instructed to generate false responses, whereas the primary objective is to identify instances in which models produce hallucinations when prompted to provide the correct answer. Hence, the proposed benchmark may not provide valuable information in evaluating actual detection models.
- The instructions used for chain of thought reasoning are not mentioned. For example, what constitutes the exemplars and how are the reasoning steps designed? What kind of reasoning do models provide for new examples? Without this information, it is not convincing that CoT performs worse.

**Reproducibility:**

3: Could reproduce the results with some difficulty. The settings of parameters are underspecified or subjectively determined; the training/evaluation data are not widely available.

**Reviewer Confidence:**

3: Pretty sure, but there's a chance I missed something. Although I have a good feel for this area in general, I did not carefully check the paper's details, e.g., the math, experimental design, or novelty.

---

> ### Author Rebuttal · Authors · 2023-08-29
>
> **Question 1: The datasets do not appear to fully align with the task at hand, specifically regarding the detection of hallucinations. Models have been intentionally instructed to generate false responses, whereas the primary objective is to identify instances in which models produce hallucinations when prompted to provide the correct answer. Hence, the proposed benchmark may not provide valuable information in evaluating actual detection models. Could the authors clarify why the benchmark is reasonable for hallucination detection, given that models were explicitly asked to generate hallucinated answers?**
>
> **Response to Reject 1:** Thanks for your comments. Our benchmark is reasonable for hallucination detection, which can be clarified from two aspects.
>
> First, the primary objective of hallucination detection is to differentiate hallucinations from correct instances. The correct instances can be readily acquired from existing datasets, whereas hallucinations should and must be sampled from the output distribution of LLMs. Therefore, it becomes both natural and reasonable to prompt LLMs to generate hallucinations as the negative instances in our benchmark. We would like to mention that this construction process (Figure 1 and Table 2) is not intended to build a model to detect hallucinations, but to produce realistic hallucinations that LLMs will likely generate. Our benchmark also includes correct and hallucination instances really generated by ChatGPT (5000 query and response pairs), which is annotated by human labelers. If an LLM can recognize the hallucinated answers, or distinguish them from the correct answers, then we can say that the LLM has some ability to detect hallucinations. This is the first benchmark of this sort.
>
> Second, although we can hire human writers to fabricate hallucinations, it is labor-intensive and may contain biases from humans. Existing powerful LLMs such as ChatGPT and GPT-4 have shown superhuman performance in many tasks and they have been widely used to generate synthesized instructions [1][2]. Therefore, instructing LLMs may generate better and plausible hallucinations, and this method has been adopted in previous literatures [3][4]. During the construction process of our benchmark, we instructed an LLM to generate hallucinated answers that are close to real hallucination patterns LLMs usually generate, by carefully designing the instructions from two aspects, i.e., hallucination patterns and demonstrations (described in Line 152 Instruction Design).  The hallucination patterns are task-specific and reflect the commonly present schema where LLMs produce hallucinations. These patterns can guarantee that the types of synthesized hallucinations are the same as in real task LLMs will produce. And the in-context hallucination demonstrations can further help LLMs understand these hallucination patterns and synthesize more accurate hallucinated examples according to the provided demonstrations.
>
> **Question 2: The instructions used for chain of thought reasoning are not mentioned. For example, what constitutes the exemplars and how are the reasoning steps designed? What kind of reasoning do models provide for new examples? Without this information, it is not convincing that CoT performs worse.**
>
> **Response to Reject 2:** Thanks for your comments. The instructions and exemplars (or demonstrations) used in CoT are similar to those in the main detection experiments (shown in Table 13, 14, and 15). The only difference is that we add the reasoning step before the final judgement, which is a common form of CoT and has been used and shown effective in many previous studies [5][6][7], so we did not provide full details about this part for simplicity. We will add these details in later versions. We show an illustrative example below.
>
> For example, in Table 13, the original instruction for the first example is:
>
> **#Question#:** What U.S Highway gives access to Zilpo Road, and is also known as Midland Trail?
>
> **#Answer#:** U.S Highway 60
>
> **#Your Judgement#:** No
>
> In CoT, the instruction with reasoning steps slightly changes to:
>
> **#Question#:** What U.S Highway gives access to Zilpo Road, and is also known as Midland Trail?
>
> **#Answer#:** U.S Highway 60
>
> **#Your Judgement#:** **According to the question, the answer must provide the name of the U.S Highway that gives access to Zilpo Road and is also known as Midland Trail. The correct answer is U.S Highway 60.** So the judgement is No.
>
> We can see that the only difference is that we add the explanatory reasoning before making the final judgement, which is a common form of CoT and has been used in many previous studies. Although CoT performs worse in QA and dialogue, it achieves better results in the summarization and general tasks. In QA and dialogue which involve much knowledge, adding chain-of-thought before output might interfere with the final judgement. The different characteristics of these tasks make the performance of CoT varying. We have explained the reasons in Line 401-412.
>
> **References:**
>
> [1] Wang, Yizhong, Yeganeh Kordi, Swaroop Mishra, Alisa Liu, Noah A. Smith, Daniel Khashabi, and Hannaneh Hajishirzi. "Self-instruct: Aligning language model with self generated instructions." arXiv preprint arXiv:2212.10560 (2022).
>
> [2] Xu, Can, Qingfeng Sun, Kai Zheng, Xiubo Geng, Pu Zhao, Jiazhan Feng, Chongyang Tao, and Daxin Jiang. "Wizardlm: Empowering large language models to follow complex instructions." arXiv preprint arXiv:2304.12244 (2023).
>
> [3] Chen, Anthony, Panupong Pasupat, Sameer Singh, Hongrae Lee, and Kelvin Guu. "PURR: Efficiently Editing Language Model Hallucinations by Denoising Language Model Corruptions." arXiv preprint arXiv:2305.14908 (2023).
>
> [4] Mündler, Niels, Jingxuan He, Slobodan Jenko, and Martin Vechev. "Self-contradictory Hallucinations of Large Language Models: Evaluation, Detection and Mitigation." arXiv preprint arXiv:2305.15852 (2023).
>
> [5] Wei, Jason, Xuezhi Wang, Dale Schuurmans, Maarten Bosma, Fei Xia, Ed Chi, Quoc V. Le, and Denny Zhou. "Chain-of-thought prompting elicits reasoning in large language models." Advances in Neural Information Processing Systems 35 (2022): 24824-24837.
>
> [6] Wang, Xuezhi, Jason Wei, Dale Schuurmans, Quoc Le, Ed Chi, Sharan Narang, Aakanksha Chowdhery, and Denny Zhou. "Self-consistency improves chain of thought reasoning in language models." arXiv preprint arXiv:2203.11171 (2022).
>
> [7] Kojima, Takeshi, Shixiang Shane Gu, Machel Reid, Yutaka Matsuo, and Yusuke Iwasawa. "Large language models are zero-shot reasoners." Advances in neural information processing systems 35 (2022): 22199-22213.

---

### Meta-Review · Area_Chair_Uyd6 · 2023-09-12

**Recommendation:** 5

**Metareview:**

Hallucinations are a major concern in text generated by current generative models. The paper proposes a benchmark for evaluating hallucinations. The dataset contains hallucinations in response to queries from HotpotQA, OpenDialKG, CNN-Dailyail). The core claim of the paper is that current instruction-tuned models do not reliably detect hallucinations, but that knowledge retrieval and CoT reasoning can help to improve the results.
The dataset is large, including 35 000 samples. The paper contains an analysis of what kinds of hallucinations occur.

The authors engaged in in-depth discussion with reviewers 1 and 3, during which many of their concerns could be addressed; the discussion resulted in several new analyses and improvements to presentation which can be incorporated in the final paper.

---

### Decision · Program_Chairs · 2023-10-07

**Decision:**

Accept-Main

**Comment:**

Hallucinations are a major concern in text generated by current generative models. The paper proposes a benchmark for evaluating hallucinations. The dataset contains hallucinations in response to queries from HotpotQA, OpenDialKG, CNN-Dailyail). The core claim of the paper is that current instruction-tuned models do not reliably detect hallucinations, but that knowledge retrieval and CoT reasoning can help to improve the results.
The dataset is large, including 35 000 samples. The paper contains an analysis of what kinds of hallucinations occur.

The authors engaged in in-depth discussion with reviewers 1 and 3, during which many of their concerns could be addressed; the discussion resulted in several new analyses and improvements to presentation which can be incorporated in the final paper.